# Assessment of Neurotoxic Effects of Oxycodone and Naloxone in SH-SY5Y Cell Line

**DOI:** 10.3390/ijms24021424

**Published:** 2023-01-11

**Authors:** Luíza Siqueira Lima, Nayara de Souza da Costa, Maria Eduarda Andrade Galiciolli, Meire Ellen Pereira, William Almeida, Marta Margarete Cestari, Pablo Andrei Nogara, Ana Carolina Irioda, Cláudia Sirlene Oliveira

**Affiliations:** 1Instituto de Pesquisa Pelé Pequeno Príncipe, Curitiba 80250-060, Brazil; 2Faculdades Pequeno Príncipe, Curitiba 80230-020, Brazil; 3Programa de Pós-Graduação em Ecologia e Conservação, Campus Politécnico, Universidade Federal do Paraná, Curitiba 81530-000, Brazil; 4Instituto Federal de Educação Ciência e Tecnologia Farroupilha (IFFAR), Campus Santo Augusto, Santo Augusto 95174-279, Brazil

**Keywords:** opioids, molecular docking, antioxidant enzymes

## Abstract

Opioid drugs have analgesic properties used to treat chronic and post-surgical pain due to descending pain modulation. The use of opioids is often associated with adverse effects or clinical issues. This study aimed to evaluate the toxicity of opioids by exposing the neuroblastoma cell line (SH-SY5Y) to 0, 1, 10, and 100 µM oxycodone and naloxone for 24 h. Analyses were carried out to evaluate cell cytotoxicity, identification of cell death, DNA damage, superoxide dismutase (SOD), glutathione S-transferase (GST), and acetylcholinesterase (AChE) activities, in addition to molecular docking. Oxycodone and naloxone exposure did not alter the SH-SY5Y cell viability. The exposure to 100 µM oxycodone and naloxone significantly increased the cells’ DNA damage score compared to the control group. Naloxone exposure significantly inhibited AChE, GST, and SOD activities, while oxycodone did not alter these enzymes’ activities. Molecular docking showed that naloxone and oxycodone interact with different amino acids in the studied enzymes, which may explain the differences in enzymatic inhibition. Naloxone altered the antioxidant defenses of SH-SY5Y cells, which may have caused DNA damage 24 h after the exposure. On the other hand, more studies are necessary to explain how oxycodone causes DNA damage.

## 1. Introduction

Opioids have been used for thousands of years to treat pain. Opium extracted from poppy seeds is the oldest opioid used in medicine, described in the third century BC [1]. Opium was responsible, in the second half of the 19th century, for the Opium War, an international conflict between China and England, because of the ban on opium consumption by China [2].

Opioids are first-pass metabolism drugs. These drugs can be administered orally, intrathecally, intravenously, subcutaneously, or epidurally [3]. Generally, opioids are metabolized in the liver and eliminated by the kidneys [4]. For instance, naloxone is deactivated in the liver 30–60 min after intravenous administration, and the inactive metabolite (6-beta-naloxol) is excreted by the urine [5]. Likewise, oxycodone is biotransformed into noxicodone (inactive metabolite) and oxymorphine (active metabolite) in the liver [6].

Among the recommended opioids sold worldwide, fentanyl, oxycodone (OxyContin^®^), hydrocodone (Vicodin^®^), codeine, and morphine stand out [7]. Although the Opioid Prescription Guideline recommends the use of opioids for limited periods, chronic pain patients make use of these drugs for long periods [8,9]. Adverse reactions after opioid treatment are usually mood alterations, psychological and psychosocial changes, and physical dependence [10,11]. Neurotoxic effects can be observed with opioid chronic use, such as cognitive dysfunction, neurodegeneration, neuroinflammation, brain abscesses, ischemic strokes, embolism, aneurysms, and leukoencephalopathy [12,13]. In 2020, 91,799 drug overdose deaths occurred in the United States, and opioids were involved in 68,630 of these deaths (74.8% of all drugs) [14].

The in vitro acute toxic effect of oxycodone and naloxone is underexplored. Due to the increase in opioid prescriptions, it is pivotal to evaluate the cellular toxicity of these drugs. Therefore, this study aimed to evaluate the toxic effects of oxycodone and naloxone in a neuroblastoma cell line (SH-SY5Y); thus, analyses were carried out to evaluate cell viability, genotoxicity, and the activity of antioxidant enzymes, in addition to molecular docking.

## 2. Results

### 2.1. Cytotoxicity

The cytotoxicity of the neuroblastoma cell line (SH-SY5Y) exposed to oxycodone and naloxone is shown in Figure 1A,B, respectively. The Kruskal–Wallis test showed no treatment effect. In fact, neither opioid drug induced a statistically significant change in cell viability. This test was also performed with the oxycodone vehicle (methanol 3.3%), which did not alter the cell viability.

### 2.2. Identification of Apoptotic and Necrotic Cells

The analysis of death by apoptosis and/or necrosis of the SH-SY5Y cell line exposed to oxycodone or naloxone is shown in Figure 2A–H. The Kruskal–Wallis test showed no effect of the treatment. SH-SY5Y cells exposed to oxycodone or naloxone did not have apoptotic or necrotic processes. The test was also performed with the oxycodone vehicle (methanol 3.3%), which did not induce an apoptotic or necrotic process.

### 2.3. DNA Damage

The analysis of the genotoxicity assay performed on the SH-SY5Y cell line exposed to oxycodone or naloxone is shown in Figure 3A,B. The Kruskal–Wallis test demonstrated the effect of oxycodone (H(5) = 8.744; *p* = 0.0064) and naloxone (H(4) = 8.016; *p* = 0.0457) exposure. The exposure to 100 µM oxycodone and naloxone caused a statistically significant increase in the score of DNA damage when compared to the nonexposed cells. The test was also performed with the oxycodone vehicle (methanol 3.3%), which did not cause DNA damage. The positive control, methyl methane sulfonate, caused a DNA damage score of 341.3, demonstrating the reliability of the test.

### 2.4. Biochemical Analysis

The biochemical analyses performed on the SH-SY5Y cell line exposed to oxycodone or naloxone are shown in Figure 4A–F. The tests were also performed with the oxycodone vehicle control (methanol 3.3%), which did not alter enzyme activities. One-way ANOVA revealed no effect of oxycodone exposure on acetylcholinesterase (AChE) activity (Figure 4A). On the other hand, one-way ANOVA revealed the effect of naloxone exposure (Figure 4B) on AChE activity (F(3,8) = 5.641; *p* = 0.0225). SH-SY5Y cells exposed to 100 µM naloxone had a statistically significant AChE activity inhibition (85%) when compared to nonexposed cells. Although 1 and 10 µM naloxone inhibited the AChE activity by ~40%, there was no statistically significant difference compared to control cells.

One-way ANOVA showed no effect of oxycodone treatment on glutathione S-transferase (GST) activity (Figure 4C). Albeit not statistically significant, the exposure to oxycodone (1, 10, and 100 µM) inhibited the GST activity by ~40% compared to control cells. On the other hand, the one-way ANOVA showed the effect of naloxone exposure (Figure 4D) on GST activity (F(3,4) = 7.364; *p* = 0.0417). The exposure to 100 µM naloxone induced a statistically significant inhibition of GST activity (~80%) compared to control cells. Although 1 and 10 µM naloxone inhibited the GST activity by 55%, there was no statistically significant difference compared to control cells.

One-way ANOVA showed no effect of oxycodone treatment on superoxide dismutase (SOD) activity (Figure 4E). Although not statistically significant, the exposure to oxycodone (1, 10, and 100 µM) caused approximately 29% SOD activity inhibition compared to control cells. The one-way ANOVA showed the effect of naloxone exposure (Figure 4D) on SOD activity (F(3,7) = 8.528; *p* = 0.0098). The exposure to naloxone (1, 10, and 100 µM) induced a statistically significant inhibition of SOD activity compared to control cells.

### 2.5. Molecular Docking

To verify the binding pose and interactions among oxycodone and naloxone with AChE, GST, and SOD enzymes, molecular docking simulations were performed (Figure 5). The predicted binding free energy (ΔG) suggested that the interactions were thermodynamically favorable, i.e., the compounds can interact in the enzyme active sites. The ΔG of oxycodone with the AChE, GST, and SOD was −7.1, −6.7, and −6.3 kcal/mol, respectively. Similarly, the ΔG of naloxone was −7.1 (AChE), −6.5 (GST), and −6.1 (SOD) kcal/mol. Despite the structural similarity between oxycodone and naloxone, the molecules showed different binding poses with the studied enzymes, suggesting that the phenol and allylamine groups from naloxone and the ether and methylamine moieties from oxycodone could interfere in the molecules’ conformation and interactions.

## 3. Discussion

The analgesic ladder of the World Health Organization determines the most appropriate form of opioid use, staggering the type of pain and opioid of choice [15]. However, opioids are frequently used incorrectly either by self-medication or by inappropriate drug prescription. Moon and Chun [16] reported three clinical cases of patients who used fentanyl without a correct prescription. The inadequate use of opioids leads to side-effects, such as alterations in pyramidal physiology [17], apnea, hypoxia, respiratory depression [11,18], constipation, nausea, vomiting, and reflux [19,20]. In this study, we observed that oxycodone and naloxone did not alter cell viability. However, exposure of cells to oxycodone and naloxone caused enzymatic inhibition and DNA damage.

The cell viability assay was carried out using 3-(4,5-dimethylthiazol-2-yl)-2,5-diphenyltetrazolium bromide (MTT). In this assay, the metabolically active cells transform the water-soluble dye MTT into an insoluble formazan [21]. In our study, there was no change in cell viability of cells exposed to 1, 10, and 100 µM oxycodone or naloxone for 24 h. In addition, neither necrosis nor apoptosis was observed after oxycodone and naloxone exposure. In contrast, Kokki et al. [22] showed that at higher concentrations (500–2000 µM), morphine and oxycodone decreased the SH-SY5Y cell viability after 48 h of exposure. Another study using drugs from the same pharmacological class showed that neuroblastoma cells exposed to tramadol (~600 µM) and tapentadol (~243 µM) for 48 h had a reduction in cell viability [23]. Moreover, Lin et al. [24] demonstrated the induction of apoptosis in 60% of cells exposed to 4000 µM of morphine for 48 h.

In this study, the exposure to naloxone inhibited the AChE activity. AChE hydrolyzes acetylcholine neurotransmitters in the cholinergic synapses; the inhibition of this enzyme leads to the interruption of acetylcholine degradation, which in turn hyperstimulates neuronal activity [25]. The inhibition of rodent AChE by opioid drugs, such as naloxone, has already been reported in the literature [25]. Motel et al. [26] suggested the use of naloxone as an adjuvant in the treatment of Alzheimer’s disease. However, inhibiting other enzymes must be considered; for instance, in this study, naloxone exposure also inhibited SOD and GST activity.

The enzyme GST is part of the antioxidant system, performing electrophilic detoxification [27]. Animal model studies demonstrated the inhibition of hepatic GST by morphine [28,29]. SOD enzyme catalyzes the degradation of superoxide anion [30]. Sadat-Shirazi et al. [31] demonstrated the inhibition of SOD activity in the prefrontal cortex of opioid abusers. SOD and GST inhibition increases reactive oxygen species and consequent oxidative stress [32,33]. As demonstrated by Ma et al. [32] and Salarian et al. [33], oxidative stress is related to addiction, dependence, and tolerance. The increase in reactive species is one of the factors linked to hyperalgesia. This factor requires ingesting a larger dose or a more potent drug to reach the desired analgesia threshold [34,35]. Therefore, the development of physiological and chemical dependence related to opioids can occur [17,36].

Even though naloxone and oxycodone have similar steric conformations, they exhibit distinct interactions with the enzymes (AChE, GST, and SOD), as demonstrated by molecular docking. Oxycodone and naloxone bind in the same region of the enzymes but interact with different amino acids. According to molecular docking, naloxone has more stable interactions. This may contribute to the different results of AChE, SOD, and GST inhibition observed in this study. Oxycodone and naloxone bind in the AChE peripheral anionic site (PAS). While oxycodone interacts with Trp286 (hydrophobic interaction) and Tyr341 (H-bonds) residues, naloxone interacts with Ser293 (H-bond) and Trp286 (electrostatic interaction). These types of interactions were observed with other AChE inhibitors [37]. Regarding GST, oxycodone and naloxone interact in the enzyme hydrophobic substrate-binding site (H-site). While oxycodone interacts with Val35 (hydrophobic interaction), Tyr108 (H-bond), and Gly205 (H-bond) residues, naloxone interacts with Val10 and Phe8 (hydrophobic interaction). In the SOD, the compounds bind near the zinc site. Oxycodone interacts with the Pro62 (hydrophobic interaction), Asn65 (H-bond), and His63 (H-bond) residues, and naloxone interacts with Lys136 (hydrophobic and electrostatic interactions).

The significant inhibition of the antioxidant system by naloxone may explain the DNA damage observed in this study; however, more studies are necessary to explain the oxycodone causing DNA damage. Sandoval-Sierra et al. [38] observed an increase in DNA methylation after 24 h opioid treatment. Another study by Tsujikawa et al. [39] demonstrated that DNA damage is a likely mechanism for morphine-induced P53 upregulation, probably through activation of the kappa-opioid receptor, which might lead to immune suppression.

In addition to the differences pointed out above, oxycodone and naloxone have different physiological action mechanisms. Oxycodone has an agonist action; in this way, the drug inhibits the pain, via the descending pathway, blocking the nociception [22]. On the other hand, naloxone has an antagonist action, inhibiting the agonists’ action, binding in the receptor, and displacing the drug that performs agonist action [40]. The different actions of these two drugs can activate distinct pathways in the cells, which may explain the results found in this work; however, more studies are needed to better explain these alterations.

## 4. Materials and Methods

### 4.1. Neuroblastoma Cell Line

The cell line SH-SY5Y (neuroblastoma) was acquired at the cell bank of Rio de Janeiro, Brazil. SH-SY5Ycells are neuroblastic-type cells, i.e., are characterized as immature nerve cells [41].

### 4.2. Cell Cultivation

Cells were cultured in Dulbecco’s Modified Eagle’s Medium/Nutrient Mixture F-12 (DMEM-F12) (Sigma-Aldrich^®^—USA) supplemented with 10% fetal bovine serum (FBS) (Sigma-Aldrich^®^—USA) and 1% penicillin/streptomycin (Sigma-Aldrich^®^—USA). SH-SY5Y cells were kept in a 75 cm^2^ culture flask at 37 °C in a humidified atmosphere with 5% CO_2_. The medium was replaced every 2–3 days.

### 4.3. Oxycodone and Naloxone Exposure

SH-SY5Y cells were plated into six- and 96-well plates at a density of 1 × 10^5^, 2.5 × 10^5^, or 1 × 10^6^ cells/well depending on the analysis performed. After 24 h, the complete DMEM-F12 medium was replaced by 0, 1, 10, and 100 µM oxycodone (Sigma-Aldrich^®^—St. Louis, MO, USA) or naloxone (Cayman Chemical Company^®^—Ann Arbor, MI, USA) for 24 h. Cells were also exposed to 3.3% methanol the oxycodone vehicle. Naloxone was prepared in DMEM-F12 medium.

### 4.4. Cytotoxicity Assay

After oxycodone and naloxone exposure, the cell medium was replaced by a 3-(4,5-dimethylthiazol-2-yl)-2,5-diphenyltetrazolium bromide (MTT, 1 mg/mL of FBS-free DMEM-F12) (Invitrogen^®^—Waltham, MA, USA) solution and the plates were incubated at 37 °C. After 3 h of incubation, the MTT solution was replaced by 100 µL of dimethyl sulfoxide (DMSO; Êxodo Cientifica^®^—Sumaré, SP, Brazil) to dissolve the formazan crystals. Absorbance was measured spectrophotometrically at a wavelength of 595 nm [42]. Results were expressed as percentages of the control.

### 4.5. Identification of Apoptotic and Necrotic Cells

After oxycodone and naloxone exposure, the cell medium was collected, and the cells were washed twice with phosphate-buffered saline (PBS, Sigma-Aldrich^®^—St. Louis, MO, USA) and harvested using trypsin (Sigma-Aldrich^®^—St. Louis, MO, USA). The cell suspensions were centrifuged at 446× *g* to remove the culture medium. After centrifugation, the cell pellets were resuspended with 250 µL of binding buffer (BD Biosciences^®^—New Jersey, NJ, USA), to which 3 µL of FITC-conjugated Annexin V (Invitrogen^®^—USA) and 5 µL of 7 aminoactinomycin D (Invitrogen^®^— Waltham, Massachusetts, USA) were added, and the cell suspensions were incubated in the dark for 15 min. A FACS Canto II flow cytometer (Becton Dickinson—Franklin Lakes, NJ, USA) with FITC and PERCP channels was used to evaluate the cells. Analyses were performed using Infinicyt software version 6.0. The results were expressed as percentages of the control [43].

### 4.6. DNA Damage

The DNA damage was analyzed using the alkaline comet assay according to Singh et al. [44] with modifications by Ferraro et al. [45]. After oxycodone and naloxone, the cell medium was collected, and the wells were washed two times with PBS and harvested by trypsin. All samples were submitted to viability analysis by the Trypan Blue (Sigma-Aldrich^®^—St. Louis, MO, USA) dye exclusion technique, and only samples with more than 80% viability were used in the comet assay. The cells were resuspended in low-melting-point agarose (Kasvi^®^—São José dos Pinhais, PR, Brazil) (0.5% *w*/*v* in PBS) and then deposited on microscope slides precoated with 1.5% agarose. After solidification under refrigeration at 4 °C for 10 min, the slides were immersed in freshly prepared lysis solution (2.5 M NaCl (Dinâmica^®^— Indaiatuba, SP, Brazil), 100 mM ethane-1,2-diyldinitrilo tetra acetic acid (EDTA; Neon^®^—Suzano, SP, Brazil), 10 mM Tris-HCl (Synth^®^—São Paulo, SP, Brazil), 1% N-lauryl sarcosinate (Synth^®^—São Paulo, SP, Brazil), 1% Triton X-100 (Synth^®^—São Paulo, SP, Brazil), and 10% DMSO) overnight. The slides were then transferred to a horizontal electrophoresis vat, filled with freshly prepared buffer solution (200 mM EDTA and 10 M NaOH (Dinâmica^®^—Indaiatuba, SP, Brazil), pH > 13) at 4 °C for 25 min for nucleoid unfolding, and then electrophoresed at 300 mA and 1 V/cm for another 25 min. After the end of this step, the slides were neutralized with Tris-HCl buffer solution (4.85%, pH 7.5) in three baths of 5 min each, and finally fixed in 100% ethanol (Alphatec^®^—Carlsbad, CA, USA). The DNA damage was analyzed using an epifluorescence microscope at 400× magnification after staining the slides with 40 µL of ethidium bromide (Sigma-Aldrich^®^—St. Louis, MO, USA) solution (0.02 g/mL). A total of 100 nucleoids per slide were visually evaluated using the “tail” size as a parameter, assuming values from 0 to 4 (0: no apparent damage and 4: maximum apparent damage). A total score was calculated from the 100 nucleoids, which assumes a value spectrum ranging from 0 to 400 [46]. To demonstrate the reliability of the test, positive control was performed with 0.5 mM methyl methane sulfonate (Sigma-Aldrich^®^—St. Louis, MO, USA).

### 4.7. Biochemical Analyses

#### 4.7.1. Sample Preparation

After oxycodone and naloxone exposure, the cell medium was collected, and the cells were washed two times with PBS and harvested by trypsin. Samples were centrifuged at 1200 rpm, the supernatant was discarded, and the pellet was washed two times with PBS. The samples were resuspended in PBS, and the aliquots were stored at −80 °C until the biochemical analyses.

#### 4.7.2. Acetylcholinesterase

AChE activity was measured as described by Ellman et al. [47]. Briefly, in a 96-well plate, 25 µL of the sample (prepared as described in Section 4.7.1), 0.75 mM 5,5-dithio-bis-(2-nitrobenzoic acid) (DTNB; Sigma-Aldrich^®^—St. Louis, MO, USA), and 10 mM acetylthiocholine (Sigma-Aldrich^®^—St. Louis, MO, USA) were added. The plate was incubated for 5 min at 37 °C, and the absorbance was measured at 405 nm once a minute for five minutes. The activity was expressed as nmol/min/mg of protein.

#### 4.7.3. Glutathione S-Transferase

The GST activity was evaluated using the method described by Keen et al. [48]. Briefly, in a 96-well plate, 20 µL of the sample (prepared as described in Section 4.7.1), 3.0 mM 1-chloro-2,4-dinitrobenzene (CDNB; Sigma-Aldrich^®^—St. Louis, MO, USA), 3.0 mM reduced glutathione (GSH; Sigma-Aldrich^®^—St. Louis, MO, USA), and 0.1 M phosphate buffer (pH 6.5) were added. Absorbance was measured at 340 nm once a minute for 4 min. The activity was expressed as nmol/min/mg of protein.

#### 4.7.4. Superoxide Dismutase

The SOD activity was performed according to the method proposed by Gao et al. [49]. In a 96-well plate, 40 µL of the sample (prepared as described in Section 4.7.1), 1 M Tris (Sigma-Aldrich^®^—USA), 5 mM EDTA (VWR Life Science^®^—Denver, CO, USA) buffer, pH 8.0, and 15 mM pyrogallic acid (Labsynth^®^—Br, Diadema, SP, Brazil) were added. After 30 min of incubation at room temperature, 1 N hydrochloric acid (Sigma-Aldrich^®^—St. Louis, MO, USA) was added to stop the reaction. The absorbance was measured spectrophotometrically at 440 nm. The results were expressed as U of SOD/mg of protein.

#### 4.7.5. Protein Quantification

Total protein was quantified as described by Bradford [50]. Briefly, 10 µL of the sample (prepared as described in Section 4.7.1) and 250 µL of Bradford reagent (Thermo Scientific^®^—Waltham, MA, USA) were pipetted into the 96-well plate. A standard curve of bovine serum albumin (BSA) (Sigma-Aldrich^®^—USA) was prepared (0–500 µg BSA/mL). The absorbance was measured spectrophotometrically at 595 nm.

### 4.8. Molecular Docking

Computational docking studies were performed using AutoDock Vina 1.1.1 [51] based on previous studies [52,53,54], with an exhaustiveness of 20. The AChE, GST, and SOD enzymes were obtained from the Protein Data Bank (PDB ID: 4EY6, 6GSS, and 2C9V, respectively), and the grid box was centered on the active site (AChE: −13.25, −46.76, 33.52 with size 30 × 30 × 30 Å; GST: 11.44, 4.81, 26.31 with size 20 × 20 × 20 Å; SOD: 22.35, −15.74, 15.49 with size 20 × 20 × 20 Å). The ligands (oxycodone and naloxone) were created using Avogadro 1.1.1 software [55] followed by semi-empirical PM6 geometry optimization using the MOPAC2016 program [56]. The amine group of the ligands was considered protonated at physiological pH, as previously reported for morphine [57]. The files were prepared for docking using AutoDock Tools 4.2 [58], in which the compounds were considered flexible and the enzymes rigid. Accelrys discovery studio visualizer [59] was used in the analysis of the results.

### 4.9. Statistical Analyses

For all tests, at least, three independent experiments were carried out. The data were statistically analyzed using the Prisma Graphpad software, version 6.0, using the Kruskal–Wallis test followed by the Dunn’s test and presented as the median ± interquartile interval (cell viability, identification of apoptotic and necrotic cells, and DNA damage) or one-way ANOVA followed by the Dunnett test and presented as the mean ± SEM (biochemical analyses). Results were considered statistically different when *p* < 0.05.

## 5. Conclusions

In conclusion, we can observe that, in addition to the chronic toxicity caused by using opioids, 24 h of exposure to naloxone or oxycodone was also harmful, causing toxic effects to the cells, such as inhibition of enzymes of the antioxidant system and DNA damage without altering cell viability. Interestingly, the naloxone exposure caused more pronounced toxic effects on the SH-SY5Y cells than oxycodone.

## Figures and Tables

**Figure 1 ijms-24-01424-f001:**
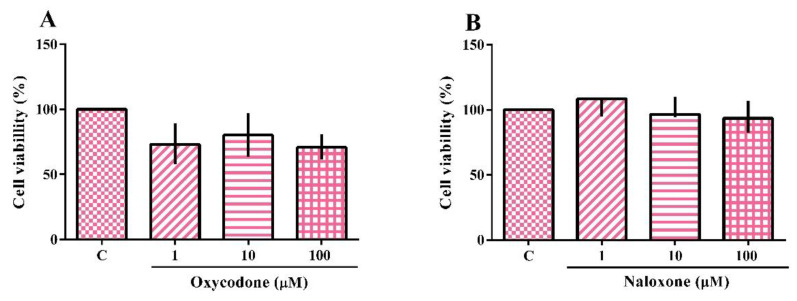
Cytotoxicity analysis of a neuroblastoma cell line (SH-SY5Y) exposed for 24 h to oxycodone (**A**) and naloxone (**B**). The results were analyzed by the Kruskal–Wallis test followed by Dunn’s post-test and presented as the median ± interquartile interval (*n* = 3–4). C: control, nonexposed cells.

**Figure 2 ijms-24-01424-f002:**
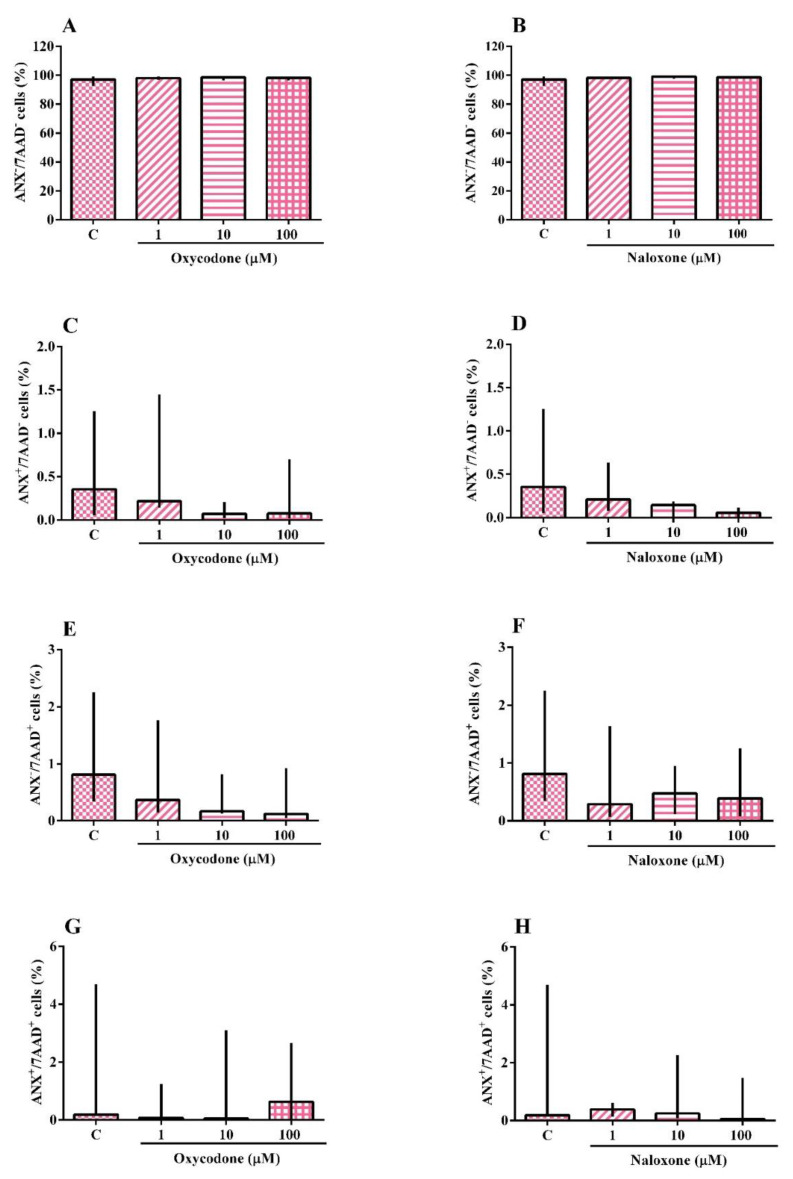
Identification of cell death by necrosis and apoptosis of a neuroblastoma cell line (SH-SY5Y) exposed for 24 h to oxycodone and naloxone. Viable cells (ANX^−^/7AAD^−^) (**A**,**B**), apoptotic cells (ANX^+^/7AAD^−^) (**C**,**D**), necrotic cells (ANX^−^/7AAD^+^) (**E**,**F**), and late apoptotic and/or necrotic cells (ANX^+^/7AAD^+^) (**G**,**H**). The results were analyzed by the Kruskal–Wallis test followed by Dunn’s post-test and presented as median ± interquartile interval (*n* = 3–5). C: control, nonexposed cells.

**Figure 3 ijms-24-01424-f003:**
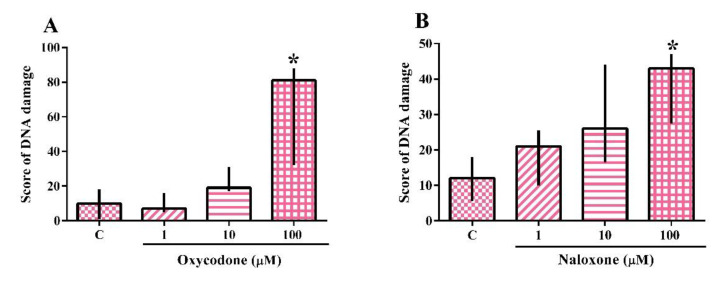
Genotoxicity analysis of a neuroblastoma cell line (SH-SY5Y) exposed for 24 h to oxycodone (**A**) and naloxone (**B**). The results were analyzed by the Kruskal–Wallis test followed by Dunn’s post-test and presented as median ± interquartile interval (*n* = 3–5). C: control, nonexposed cells. * Statistically different from C.

**Figure 4 ijms-24-01424-f004:**
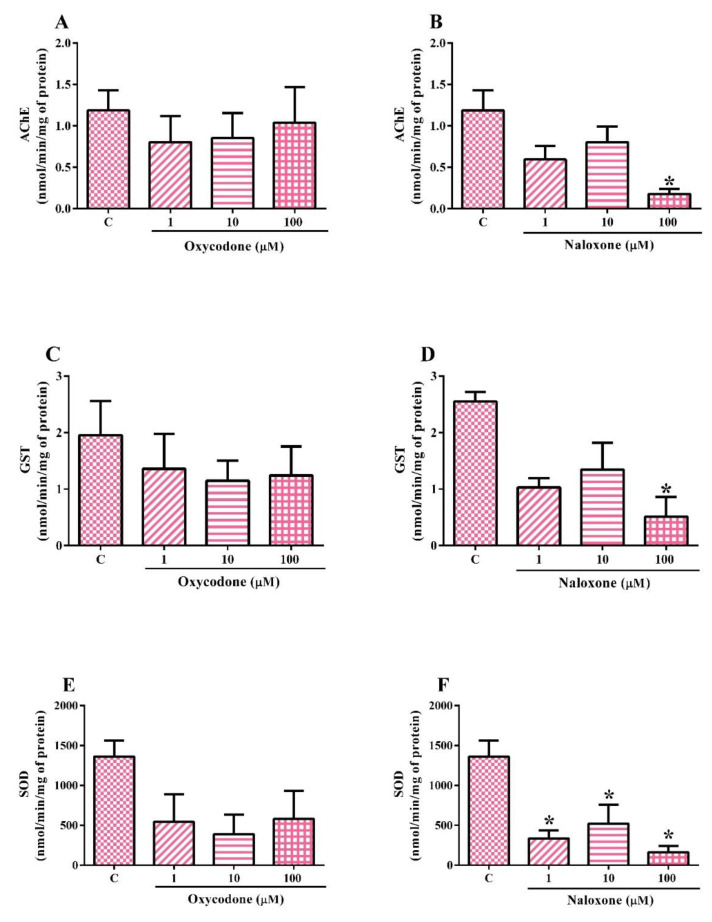
Biochemical analysis of a neuroblastoma cell line (SH-SY5Y) exposed for 24 h to oxycodone (**A**,**C**,**E**) and naloxone (**B**,**D**,**F**). The results were analyzed by one-way ANOVA followed by Dunnett’s post-test and presented as the mean ± SEM (*n* = 2–3). C: control, nonexposed cells. * Statistically different from C.

**Figure 5 ijms-24-01424-f005:**
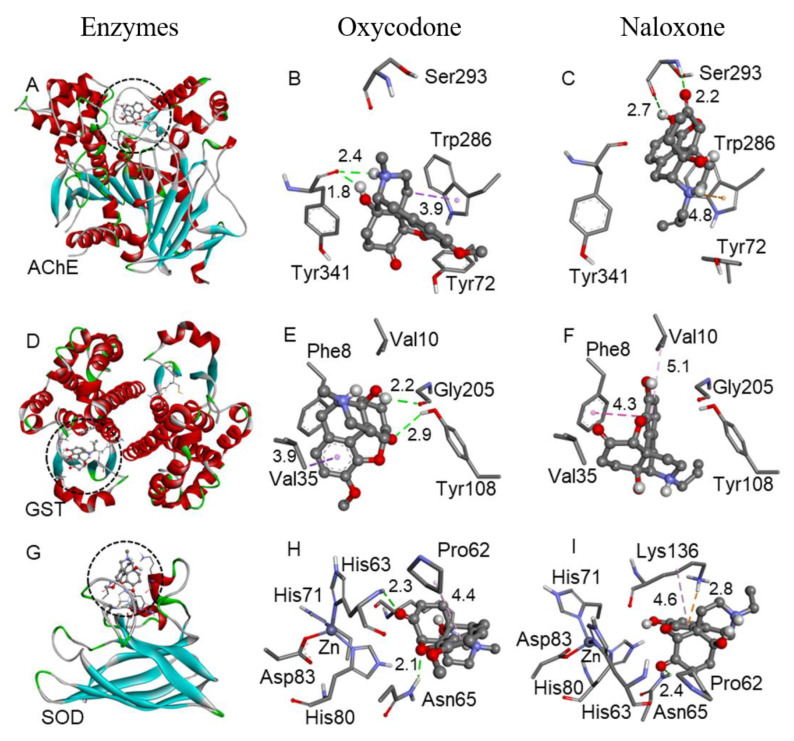
Binding poses of naloxone and oxycodone with the AChE (**A**–**C**), GST (**D**–**F**), and SOD (**G**–**I**) enzymes. Oxycodone (**B**,**E**,**H**) and naloxone (**C**,**F**,**I**) are represented by the ball-and-stick model. Only the main residues involved in the interactions are shown in the stick model. The interaction distances are in Å, with the H-bonds (green), electrostatic (orange), and hydrophobic interactions (purple) represented by dotted lines.

## Data Availability

All data generated or analyzed during this study are included in this published article.

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
