# Peer review of "Assessment of Neurotoxic Effects of Oxycodone and Naloxone in SH-SY5Y Cell Line"

_ijms, 2023, doi:10.3390/ijms24021424_

Round 1

Reviewer 1 Report

In the present work, the authors studied the acute toxicity of oxycodone and naloxone using a neuroblastoma cell line by evaluating their effect on apoptosis, oxidative stress, genotoxicity, and viability. Experimental data is also explained in the light of in silico analysis. Moreover, the authors have made a careful exposition of their experimental and computational methods and data analysis techniques. However, the authors should rewrite the conclusion section to emphasize the results. Overall this is an exciting manuscript showing the short-term effects of opioids on the cells.

Author Response

Thank you for the suggestions, Please see the attachment

Reviewer 2 Report

The manuscript "Assessment of neurotoxic effects of Oxycodone and Naloxone in SH-SY5Y cell line" describes toxicity effect of two drugs oxycodone and naloxone on neuroblastoma cell line. The viability, cell death and activity of DNA damage enzymes has been shown in the manuscript for short term period i.e. 24hrs. The manuscript discusses interesting effect on cells and their probable cause. However there are a few comments which needs to be  addressed that would further enhance the manuscript.

Comments:

1. The mechanism of DNA damage is not very clear in response to these drugs. Thus authors are suggested to perform additional test for example Modified comet assay.

2. Results of biochemical analysis shows that the enzyme activity inhibited up to some extent on exposure to oxycodone and naloxone, howevere statistically significant inhibition was observed only in some cases. Therefore, authors are suggested to Determine Intracellular Reactive Oxygen (ROS) Production in order to get confirmatory results and explore the mechanism of cytotoxicity

3. In previous studies on neurotoxicity of opioids and other drugs of same pharmacological class, cells were exposed to higher concentration of drugs for 48 h. Previous studies demonstrate that at high concentration (>200-4,000 μM) and for period >24h, opioids drugs causes decrease in cell viability and death, therefore, this study should be performed for 48h to clarify the acute toxicity of these drugs at lower concentration 1 μM, 10μM and 100μM (i.e >200 μM).

4. In general, the effect of oxycodone need little more discussion in the manuscript, mechanism etc.

5. Authors are suggested to check the combined effect of these drugs on toxicity of SH-SY5Y cells. 

6. Authors are suggested to correct grammatical errors and check the references.

Author Response

(The authors gave the same response as above.)

Round 2

Reviewer 2 Report

Authors provided logical explanation for the comments. The manuscript is revised accordingly and is recommended for publication in the journal.